# Full-Mouth Rehabilitation of a Patient with Gummy Smile—Multidisciplinary Approach: Case Report

**DOI:** 10.3390/medicina59020197

**Published:** 2023-01-19

**Authors:** Kinga Mária Jánosi, Diana Cerghizan, Florentin Daniel Berneanu, Alpár Kovács, Andrea Szász, Izabella Mureșan, Liana Georgiana Hănțoiu, Aurița Ioana Albu

**Affiliations:** 1Faculty of Dental Medicine, George Emil Palade University of Medicine, Pharmacy, Science, and Technology of Targu Mures, 38 Gh. Marinescu Str., 540142 Targu Mures, Romania; 2Private Practice, SC Maxdent Office SRL, 540501 Targu Mures, Romania

**Keywords:** oral rehabilitation, laser-assisted crown-lengthening, piezo-surgery, implants, zirconia ceramics

## Abstract

The impairment of aesthetic function leads to a decreased quality of life. An unaesthetic smile due to excessive gingival exposure demands, most of the time, a complex treatment in which the objective is the vertical reduction of the amount of exposed fixed gingiva by obtaining a complete exposure of the anatomical crown of the teeth and restoring the ideal dimensions of the biological width. This paper presents a case of a 48-year-old female patient who was unsatisfied with her aesthetics and had disturbed masticatory function due to the absence of some posterior teeth. The cone beam computed tomography was performed to evaluate the facial and dental morphology. The treatment plan included diode laser and piezo-surgery utilization for the frontal area of the upper arch and implants to restore the distal area of the lower and upper arch. Zirconia ceramic was used for the final restorations. This complex and multidisciplinary full-mouth rehabilitation lasted for two years, and the patient was pleased with the result. This case showed that a well-established treatment plan is necessary to obtain long-lasting results. The use of adequate procedures and equipment ensures a predictable result.

## 1. Introduction

Inadequate facial aesthetics due to an unaesthetic smile, especially in the case of a gummy smile, with a high smile line, can harm patients’ quality of life, even leading to psychical problems in some cases [1]. The American Academy of Periodontology (AAP) defines the gummy smile as a deformity and mucogingival condition that affects the area around the teeth [2]. The excessive gingival display is characterized by overexposure of the maxillary gingiva during smiling or speaking [3]. According to Allen, gum exposure of less than 2–3 mm can be considered attractive. An overexposure of more than three mm is known as the gummy smile [4] and is generally considered an aesthetic problem [5]. In some European countries, a gingival display up to 4 mm or more is acceptable [4]. Etiological factors related to a gummy smile can be gingival (passive eruption), skeletal (vertical maxillary excess), and muscular (upper lip hyperfunction) [6]. The high smile line and excessive gingival exposure must be considered during the treatment plan [6] because sometimes corrections are needed during full-mouth rehabilitation.

The treatment procedure depends on the diagnosis and the etiological factors.

Aesthetic crown lengthening is one of the most common surgical treatments for a gummy smile. The objective of this procedure is the vertical reduction of the amount of exposed fixed gingiva by obtaining a complete exposure of the anatomical crown of the teeth and restoring the ideal dimensions of the biological width [7]. Following crown-lengthening surgery, the biological width is restored to a minimum of 2 mm, with the epithelial attachment of 0.97 mm and connective tissue of 1.07 mm width [8].

During the development of the treatment plan, tridimensional imagistic investigations are necessary. Cone beam computed tomography (CBCT) can provide accurate information about the alveolar bone and the periodontal status of the teeth. Measurements can be performed to define the length of the anatomical crown and root, which is necessary to realize the surgery correctly [9].

The greatest desire during the surgery is good visibility, without bleeding in the working area. The diode laser’s most significant advantages are the non-bleeding operative field, tissue evaporation ability, adequate sterilization of the interventional area and minimal postoperative pain and edema [10].

Piezo-surgery offers a promising alternative to bone resection with significant benefits compared to traditional methods. It reduces the bleeding rate by 25–30% because it does not damage the soft tissues or blood vessels, ensuring a clean operating field during the intervention [11]. Its combination with the minimal flap technique significantly reduces postoperative pain and edema [12].

The gingival phenotype and the suture technique influence the evolution of the healing process after the surgery [13].

Dental implants have been considered one of the most important discoveries in dentistry in the past decades. In modern dentistry, the implant-prosthetic approach allows the treatment of partially edentulous spaces with fixed restorations, considerably improving the patient’s quality of life [14]. The implant therapy, combined with zirconia ceramic restorations, allows the rehabilitation of function and aesthetics [15].

## 2. Case Report

This case report is a full-mouth rehabilitation of a 48-year-old female patient. She wanted to improve her aesthetics, disturbed by the shape and orientation of the upper frontal teeth and the excessive visibility of the gingiva. The patient also reports difficulties in mastication due to the absence of numerous posterior teeth in the lower arch. To establish the preliminary diagnosis, intraoral examinations (Figure 1a) and a panoramic X-ray (Figure 1b) were performed.

The clinical examination revealed the presence of inadequate metal-ceramic restorations, teeth with unsatisfactory periodontal status (grade I mobility), aesthetical and functional problems. A full-mouth CBCT scan was performed for the final diagnosis and to establish the treatment plan. The treatment objective was to perform full-mouth rehabilitation and improve the smile’s aesthetics by reducing the excessive gingival displacement.

A crown-lengthening surgery was planned before the prosthodontic rehabilitation. The long-term success of future restorations is conditioned by accurately reestablishing the vertical dimension and the occlusal plane. The functional rehabilitation of the jaws needed an implant-prosthodontic approach. The treatment plan was established following the patient’s agreement, considering the principles of the Declaration of Helsinki involving humans as revised in 2013. Informed consent was obtained from the patient regarding the treatment, and written informed consent has been obtained to publish this paper.

The full-mouth rehabilitation of this case was performed for two years.

Based on the CBCT measurements, the postoperative maxillary crown/root ratio was defined (Figure 2). The right central incisor presented a short root length to support future exposure during the crown-lengthening procedure. Therefore, it was decided to extract this tooth. The extraction of periodontally compromised 17 and 15 teeth was recommended with implant-prosthodontic rehabilitation of the right posterior area.

The surgical pre-prosthetic treatment protocol combines laser therapy with the piezo-surgery to achieve a minimally invasive intervention with reduced postoperative symptomatology. Intraoral mock-ups were created to simulate and individualize future results and to guide the surgery (Figure 3).

The old restorations were removed. The surgical guide was realized after the preliminary preparation of the abutments with the vertical preparation technique. During the surgical interventions, the Optragate (Ivoclar Vivadent AG, Schaan, Principality of Liechtenstein) retractor was used, which ensured good visibility and adequate access to the working area. The Lasotronix Smart M Pro diode laser (Lasotronix Sp. z o.o., Piaseczno, Poland) was used for the guiding incisions at the gingival margin, following the cervical line of the mock-up. No elongation was performed at the right central incisor because this tooth needed to be extracted at the end of the surgery. The alveolar bone margins were removed using the Ultrasurgery US-III LED piezo-surgery device (Guilin Woodpecker Medical Instrument Co., Ltd, Guangxi, P. R. China) (Figure 4).

To obtain long-lasting results, the placement of the margins of the future restorations must be at a minimum distance of 5 mm from the alveolar bone. Therefore, this desirability was considered during the surgery (Figure 5).

After performing the surgery and extracting the right central incisor, socket preservation and bone augmentation were done to maintain the alveolar bone dimensions. Provisional restoration was made to restore the aesthetics and function temporarily. Complete healing was achieved after six months. As expected, the results obtained were stable. The gingival contour was exposed but symmetrical and satisfactory during the smile. The final preparation of the abutments was performed with a heavy chamfer finish line, and zirconia ceramic restorations were used for the prosthodontic rehabilitation (Figure 6). The bite template was used to reestablish the vertical dimension. The color of the restorations was B1 (Vita Classical shade guide).

After the extraction of teeth 15 and 17, two implants were inserted. The osteointegration of the implants can be seen in the panoramic X-ray after six months (Figure 7).

Pre-prosthetic treatments were performed on the lower arch during the upper arch healing period. The endodontic retreatments of the lower premolars were successful. The preparation of the teeth was carried out with a subgingivally placed heavy chamfer finish line. Single crown zirconia ceramic restorations and a bridge were realized, preserving tooth vitality in all abutments (Figure 8).

For the restoration of the edentulous space on the lower arch, implant therapy was applied. Two implants were inserted.

In the case of the upper arch, a screw-retained titanium-based zirconia ceramic bridge was realized after the osseointegration period. The closed impression tray technique was used (Figure 9).

A panoramic X-ray was taken to verify the osseointegration of the lower implants (after six months) (Figure 10).

Due to the lack of parallelism of the implant bodies, a cemented zirconia ceramic bridge was realized to re-establish the function on the lower arch. In this case, the open-tray technique was used for the impression (Figure 11).

The final aspect of the complex, multidisciplinary full-mouth rehabilitation is presented in Figure 12.

## 3. Discussion

Several studies have demonstrated the need for these surgical interventions to obtain an aesthetic smile, predominantly in the case of female patients [16], as in the presented case. The patient’s initial problem was the gingival smile with the visibility of unaesthetic metal-ceramic prosthetic works. In this situation, resolving the patient’s primary problems was possible only by performing full-mouth rehabilitation.

The surgical correction of the upper clinical crown:root ratio and gingival displacement was necessary before the prosthodontic approach.

According to Narayan et al., the pretreatment planning included a complex clinical evaluation regarding:The patient’s systemic health and her expectations;The evaluation of the face and smile line;The lip thickness and size;The size and shape of the teeth;The gingival biotype and the width of keratinized gingiva;The thickness and contour of the alveolar bone [17].

CBCT evaluation was performed by measuring the existent and the future crown: root ratio and the crestal bone relation to the cementoenamel junction to decide the surgery type.

In the presented case a, mock-up guided crown lengthening procedure was performed based on the diagnostic wax-up, similar to the technique described by Jurado et al. [18] in their case report. Using a precise 3D-printed surgical guide for crown lengthening can help to prevent or reduce the chance of under or over-contouring hard and soft tissues during the procedure [19].

The crown-lengthening surgery combined two modern, minimally invasive techniques (laser therapy for soft tissue remodeling and piezo-surgery for bone resection) and the conventional technique to obtain long-lasting results with minor post-interventional symptoms and reduced healing time. The methods reduced surgical chair time and operative trauma, accelerating the healing process and making the patient more comfortable. The flapless surgery was undesirable because it did not allow direct visualization of the operative field and can be challenging regarding soft tissue damage [20]. Performing a reduced flap without vertical incisions was beneficial.

The thick gingival phenotype of the patient facilitated the healing process. Three months post-operatively, stabile results were obtained, probably due to minimally invasive techniques and the favorable gingival phenotype. The recovery period, a controversial topic in the literature, can differ individually. After soft tissue remodeling, the final rehabilitation can be done after a healing period of three months [21,22]. According to Herrero et al., in the case of bone remodeling with biological width modification, the healing period must be about six months before the prosthodontic rehabilitation [23], and it is essential to define a proper distance between the finish line and the bone margin during post-surgical prosthodontic treatment [24]. In our case, the healing period was six months.

After this period, the re-preparation of the teeth was carried out with a subgingivally-placed heavy chamfer finish line at a greater distance from the bone margin than 5 mm. Zirconia ceramic single crown restorations and bridges were used for aesthetic and functional rehabilitation. Several studies have been carried out regarding the marginal adaptation of these restorations, which are superior to conventional metal-ceramics [25,26]. Proper teeth preparation and a good impression technique are essential to achieve the best results [27,28]. In the case of digital workflow during the zirconia frame’s design, the cementation space can influence the quality of the marginal adaptation. Defining the dimensions of this space must be done with caution [29]. Dittmer et al. [30] and Kohorst et al. [31] demonstrated in their studies that the successive application and firing of ceramic layers on the zirconia frame could cause marginal discrepancies, contradictory to Vigolo et al. findings [32]. The zirconia framework presents a lower occurrence of discrepancies than metal-ceramics [33]; this can contribute to obtaining long-lasting aesthetical results. The perfect marginal fit of the restorations is essential in maintaining periodontal health and ensuring the restorations’ natural appearance, especially in the frontal area.

In the literature, different recommendations can be found for the cementation of zirconia ceramic restorations on teeth and implants. Some studies recommend the adhesive cementation technique in case of poor retention of the abutments [34,35]. Other studies have shown the importance of treating the inner surfaces of zirconia restorations to achieve good adhesion after cementation [36,37]. In the presented case, the zirconia restoration internal surface was sandblasted and treated with Ivoclean (Ivoclar). The vitality of the teeth influenced our choice of adhesive material. The adhesive cementation was abandoned to avoid pulpal irritation related to etching. Resin-modified glass ionomer cement was used for the final cementation of the restorations in the case of natural teeth.

In the case of implants, the fixation method (screw or cement retained) of the restorations might not directly influence their survival rate. However, it can lead to certain complications (mainly periimplantitis) [38]. Each retention method has its indication with advantages and disadvantages [39]. According to de Brandao et al., there is no evidence of differences in the marginal bone loss around the cement and screw-retained restorations [40]. Several studies demonstrated a higher success rate of screw-retained restorations versus classically cemented ones [38,41]. Park et al. recommend choosing the appropriate fixation method depending on the implants’ parallelism and considering the occlusal relations. It is crucial in the case of the upper premolar region the possibility to place the access hole of the screw on the central fossa [42], as it was in our case.

The patient was satisfied with the obtained results, even though she still had a moderate gummy smile. The lip-repositioning surgery represents future possibilities for better aesthetical results [43], as does the injection of botulinum toxin A [44].

The patient chose a less invasive way to improve the final aesthetics in the future by using hyaluronic acid filler to make the lips look fuller and more youthful.

The limitations during the follow-up:Lack of periodical CBCT evaluation (at three months, six months, and one year)Lack of periodical periodontal evaluation using periodontal probing.

Digital planning and using a 3D-printed surgical guide can improve the expected results. A good collaboration between a multidisciplinary dental team and a facial plastic surgeon can result in even better aesthetics.

## 4. Conclusions

The crown-lengthening surgery is an efficient method to improve aesthetics in the case of a gingival smile. Laser therapy and piezo-surgery are modern methods that allow minimally invasive and efficient interventions with fast postoperative recovery. The zirconia ceramic restorations can be used to restore aesthetics and function with good results. Screw-retained restorations have a better long-term prognosis compared to cemented ones, demonstrated by the one-year follow-up Panoramic X-ray.

## Figures and Tables

**Figure 1 medicina-59-00197-f001:**
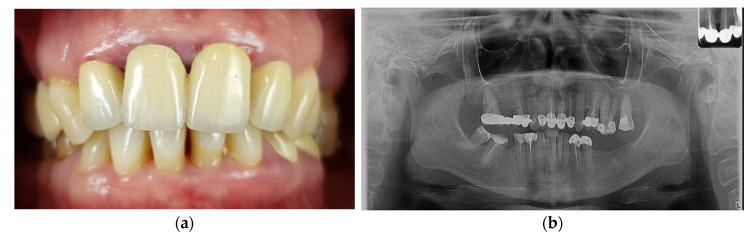
Initial situation of the patient: (**a**) Unaesthetic metal-ceramic crowns with chronic inflammation of the gingival margins and oblique interincisal line; (**b**) Initial panoramic X-ray.

**Figure 2 medicina-59-00197-f002:**
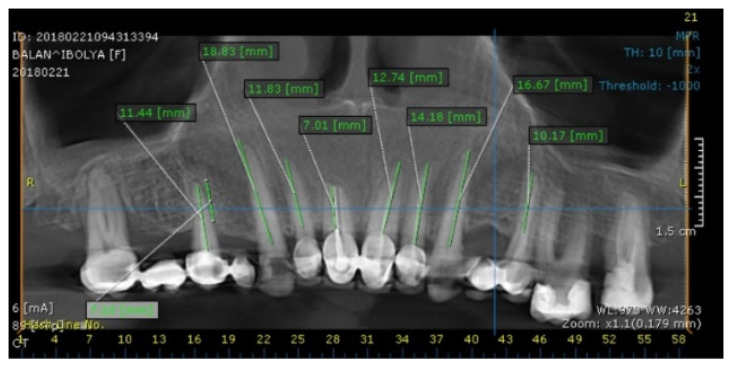
Evaluation of the periodontal status and root length of the maxillary teeth on CBCT.

**Figure 3 medicina-59-00197-f003:**
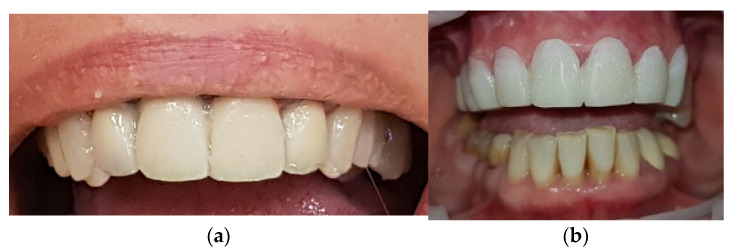
Planning the surgical treatment outcomes: (**a**) Initial mock-up; (**b**) Surgical guide.

**Figure 4 medicina-59-00197-f004:**
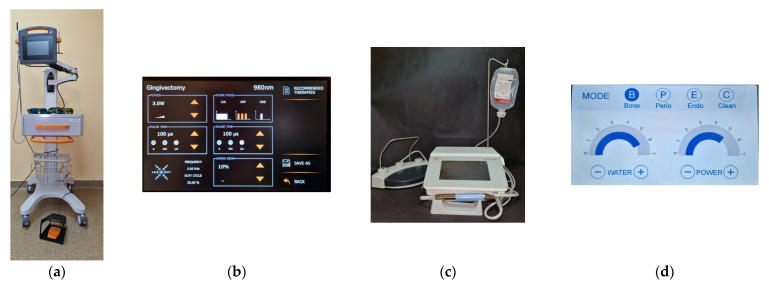
The used devices and the settings: (**a**) Lasotronix Smart M Pro laser; (**b**) The setting of the laser device for the gingivectomy; (**c**) Ultrasurgery III LED piezo-surgery device; (**d**) The setting of the piezo-surgery device.

**Figure 5 medicina-59-00197-f005:**
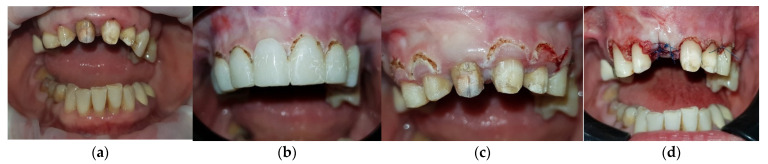
The steps of the crown-lengthening surgery: (**a**) The abutments after the removal of the old restorations and a preliminary preparation before the surgery; (**b**) Predefinition of the gingival margins using the surgical guide and the Lasotronix Smart M Pro laser; (**c**) The limits of the new gingival margins after the removal of the surgical guide; (**d**) The aspect of the prosthodontic field after the piezo and laser surgery.

**Figure 6 medicina-59-00197-f006:**
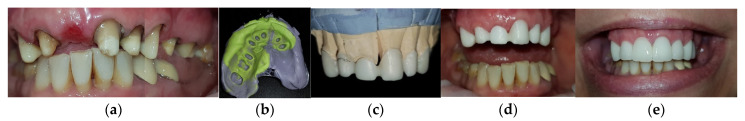
The steps of the prosthodontic rehabilitation: (**a**) Final aspect of the abutment after the healing period; (**b**) One-step impression with A silicone—Variotime Heavy Tray and Medium Flow (Zhermack); (**c**) The zirconia frame on the master cast; (**d**) The try-in of the zirconia frame; (**e**) The final restoration after cementation.

**Figure 7 medicina-59-00197-f007:**
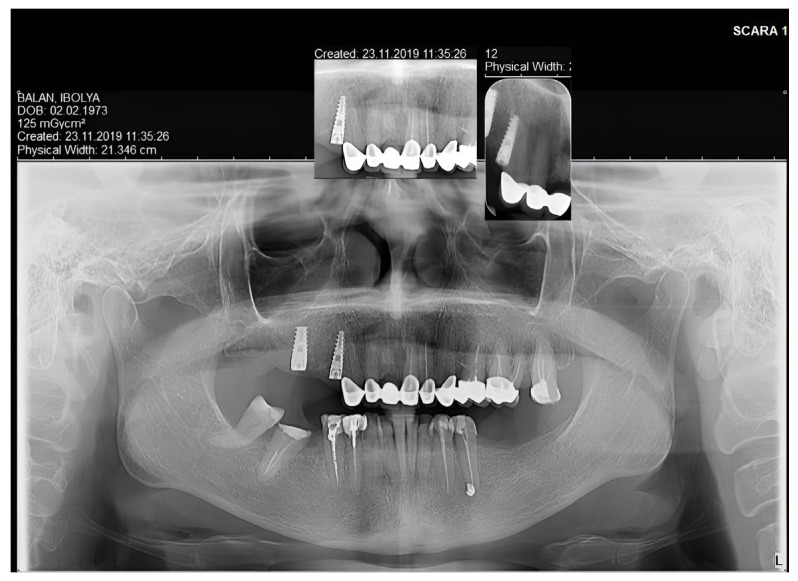
Panoramic X-ray with the osseointegrated implants and the maxillary prosthodontic rehabilitation on the natural teeth.

**Figure 8 medicina-59-00197-f008:**
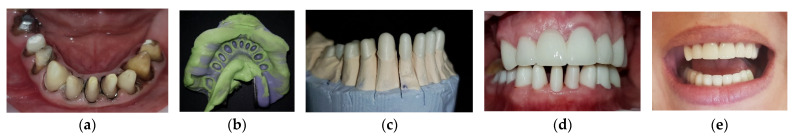
The steps of the prosthodontic rehabilitation: (**a**) The prepared abutments with the first impression cord; (**b**) One step impression with A silicone; (**c**) The zirconia frame on the master cast; (**d**) The try-in of the zirconia frame; (**e**) The final restoration after cementation.

**Figure 9 medicina-59-00197-f009:**
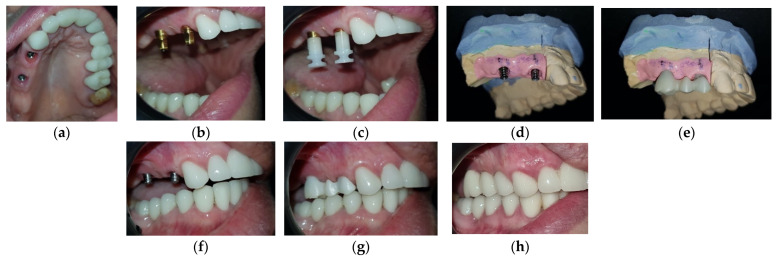
The sequences of the implant-prosthodontic therapy in the maxilla: (**a**) The emergence profile after the removal of the healing caps; (**b**) The impression copings in the mouth; (**c**) The transfer caps applied on the impression copings; (**d**) The master cast with the artificial gingiva and titanium abutments; (**e**) The zirconia frame on the master cast; (**f**) The intraoral try-in of the titanium abutments; (**g**) The intraoral try-in of the zirconia frame; (**h**) The final restoration after the intraoral fixation.

**Figure 10 medicina-59-00197-f010:**
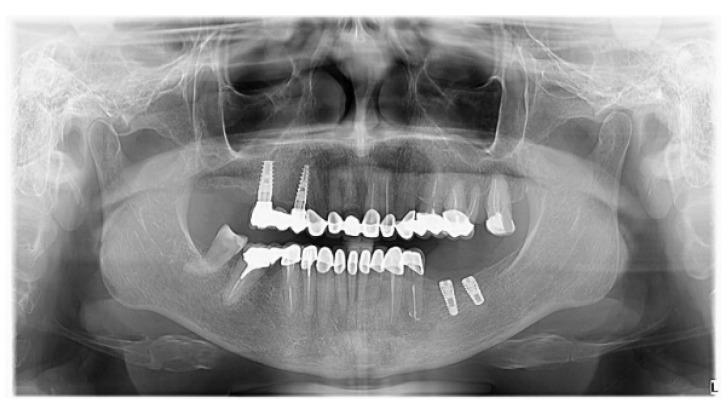
Panoramic X-ray with the osseointegrated implants on the lower arch and the good marginal adaptation of all the restorations.

**Figure 11 medicina-59-00197-f011:**
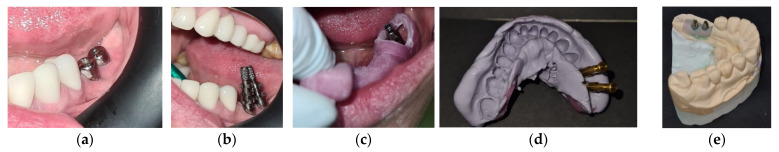
The sequences of the implant-prosthodontic therapy in the mandible: (**a**) The healing caps; (**b**) The impression copings in the mouth; (**c**) The try-in of the open tray; (**d**) The impression with the technical analogs (**e**) The prepared abutments and the artificial gingiva on the master cast; (**f**) The zirconia framework on the cast.; (**g**) The final restoration on the cast; (**h**) The try-in of the zirconia frame; (**i**) Closing the hole on the abutments; (**j**) The final restoration after the cementation.

**Figure 12 medicina-59-00197-f012:**
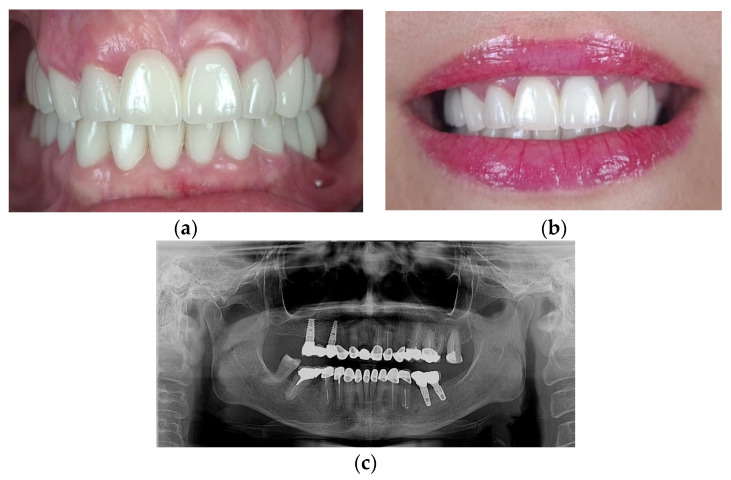
The final result of the full-mouth rehabilitation: (**a**) Intraoral aspect of the restorations; (**b**) The improved smile with minimal gingival display; (**c**) Panoramic X-ray after one year.

## Data Availability

The dataset analyzed during this case report are available from the first author on request.

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
