# Peer review of "Full-Mouth Rehabilitation of a Patient with Gummy Smile—Multidisciplinary Approach: Case Report"

_medicina, 2023, doi:10.3390/medicina59020197_

Round 1

Reviewer 1 Report

This is an interesting manuscript that reports a case that reports a case of full-mouth rehabilitation of a patient with a gummy smile. The findings may be helpful in planning dental treatments. The topic investigated is certainly of interest, however, the following points are to be addressed:

Abstract: The sentence “The treatment plan involves a cone beam computed tomography evaluation” seems wrong. The cone beam computed tomography was performed to evaluate the facial and dental morphology. The treatment plan included…”

Introduction: The text is confusing. The authors list a series of information related to the clinical case, but the sentences do not present a fluidity in the reading and the ideas are disconnected. I suggest preparing thematic paragraphs: 1st paragraph (Description of the gummy smile and aesthetic repercussions and on the quality of life of patients). 2nd paragraph (important procedures in planning the clinical treatment of this type of clinical case). 3rd paragraph (description, advantages/disadvantages of the clinical-surgical interventions used in the present case). 4th paragraph (differentials and purpose of this case report). Better articulate the central ideas of the paragraphs.

Case report: Please, change "lateral teeth" to “posterior teeth" (Page 2, line 60). In figure 1, it would be interesting to include a photo with a smiling mouth. In figure 1, it would be interesting to include an initial photo with a smiling mouth. Change 1.7 and 1.5 teeth to 17 and 15 teeth throughout the text.

Discussion: The use of a single paragraph in the discussion section is confusing. Use thematic paragraphs in the discussion relating to the scientific literature: main findings of this case report; the importance of the careful initial evaluation and what was done in the present case; important points in the clinical surgical interventions selected in the reported treatment; limitations observed during the follow-up of the clinical case. New directions and future recommendations based on case findings. In addition, some sentences seem dispensable, for example: "The phonetical tests can be used to evaluate the dimensional accuracy of the restorations". Was any phonetic assessment carried out in the case? This sentence appears out of place in the text.

Author Response

Happy New Year!

Reviewer 2 Report

The aim of the study is to present a case of a 48-years-old female patient with gummy smile.

The presented treatment plan involves a cone beam computed tomography evaluation, diode laser, and piezo-surgery utilization for the frontal area of the upper arch and implants in order to restore the distal area of the lower and upper arch.

The work highlighted, that presented complex and multidisciplinary full-mouth rehabilitation lasted for two years, and the patient was pleased with the result.

This is a well-written paper. The manuscript is well structured. Methods and data are clearly shown. All the above-mentioned factors are described clearly. 

However, I have several suggestions. 

 1. Introduction

- Line 28-35: 

– please provide more information about gummy smile including smile line.

2. Discussion:
- Line 188-189: The recovery period, a controversial topic in the literature, can differ individually. After soft tissue remodeling, the final rehabilitation can be done after a healing period of three months.

- please provide citations
- Line 183-184: In our case a mock-up guided crown lengthening procedure was performed, based on the diagnostic wax-up, similar to the technique used by Jurado et al. [15] in their case report. 

– please remove the bold

Overall:
In general, the work is interesting and can contribute to the literature. I hope my suggestions will help improve this work.

Author Response

Happy New Year!

Reviewer 3 Report

This is a nice described case.  The initial situtation showed long crowns and the crown length was further enlarged by the periodontal surgery. Why, please provide a sound reasoning.  Furthermore, the patient will still have a gummy smile, please show comparable pretreatment and posttreatment photograph to see the effect obtained. Now the pretreatment and posttreatment photographs are not comparable. This is a must for this kind of case report. 

Author Response

Happy New Year!

Round 2

Reviewer 1 Report

I have no more concerns on this manuscript. English language and style are fine/minor spell check required.

Author Response

Thank you for Your approval.

We performed an English-language spell check.

Reviewer 3 Report

The paper has improved. However, the missing link is still the initial situation. Now only pictures are shown from the mock-up stage and afterwards. These pictures are also not shown with the same smile line, therefore the true effect can not be estimated. It is up to the editor to consider this as a major flaw or not.

Author Response

Thank you for your review.

Unfortunately, we still couldn`t find the initial smiling picture.